# TSGM: A Flexible Framework for Generative Modeling of Synthetic Time Series

Alexander Nikitin[1]    Letizia Iannucci[1]    Samuel Kaski[12]

[1] Department of Computer Science, Aalto University
[2] Department of Computer Science, The University of Manchester
`alexander.nikitin@aalto.fi`

## Abstract

Time series data are essential in a wide range of machine learning (ML) applications. However, temporal data are often scarce or highly sensitive, limiting data sharing and the use of data-intensive ML methods. A possible solution to this problem is the generation of synthetic datasets that resemble real data. In this work, we introduce Time Series Generative Modeling (TSGM), an open-source framework for the generative modeling and evaluation of synthetic time series datasets. TSGM includes a broad repertoire of machine learning methods: generative models, probabilistic, simulation-based approaches, and augmentation techniques. The framework enables users to evaluate the quality of the produced data from different angles: similarity, downstream effectiveness, predictive consistency, diversity, fairness, and privacy. TSGM is extensible and user-friendly, which allows researchers to rapidly implement their own methods and compare them in a shareable environment. The framework has been tested on open datasets and in production and proved to be beneficial in both cases. `https://github.com/AlexanderVNikitin/tsgm`

## 1   Introduction

Time series data are widely used across many applications in science and technology, including health informatics [81, 36], dynamical systems [79], weather forecasting [10, 25], and predictive maintenance [51, 43]. Machine learning methods have proven to be applicable to some of these areas, largely due to the quality of the models and the availability of datasets and benchmarks. This work focuses on the latter, developing a framework that extends available datasets with synthetic time series.

A prime example of the significance of data availability is the deep learning revolution [63] of the last decade. The creation of large, rich datasets, such as ImageNet [18], CIFAR [37], or IMDB [47] has led to the development of deep neural networks for images and texts, producing state-of-the-art results in those domains [73, 32, 21, 55]. Open datasets are now expanding beyond these areas to include large time series datasets. However, such datasets are applicable to only a small fraction of the predictive problems involving time series. For many of these problems, open datasets are either insufficiently large or lacking entirely. Several factors limit data availability, including the nature of the problem (e.g., epidemiological data across countries) and privacy concerns [7, 83]. In particular, training foundation time series models requires synthetic data [2, 16]. A viable solution to these issues is the development of synthetic time series data generators.

To tackle the lack of datasets in the domain of temporal problems, various synthetic time series generation methods have been proposed, including approaches based on GANs [78, 50, 44] or VAEs [19, 41, 40]. Despite the development of many methods, the lack of a unified set of metrics for comparison and the absence of a software framework have hindered the progress and the applicability

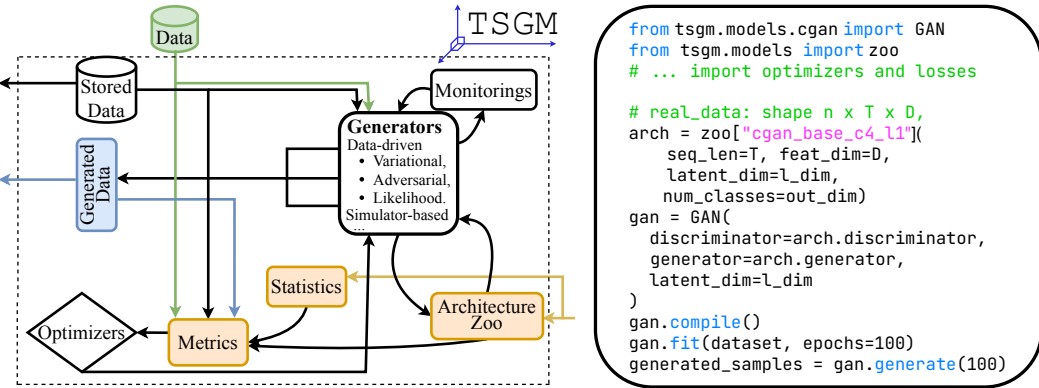

Figure 1: The architecture of TSGM. The *generators* are the core of the framework, implementing various generative methods for time series. *Architecture Zoo* provides a collection of NN architectures that can be reused at different stages of the pipeline; it can be extended by user-defined models. The *monitorings* module provides a set of routines for examining the training procedure and helps to check convergence and intermediate results. The *statistics* module implements summary statistics used by metrics. The *metrics* module evaluates the quality of the generated data and is used either during training or for the final evaluation of the generated data. The code example (right) demonstrates synthetic dataset generation with TSGM.

of these methods to real-world problems. In this work, we introduce our framework, TSGM, which advances the broader applicability and unification of synthetic time series data generation.

**Contributions.** Our main contributions are the following

- We introduce a software framework for synthetic time series dataset generation that operates within the Keras ecosystem and is extendable to TensorFlow, Torch, and Jax, providing a unified interface across different communities.

- We introduce a comprehensive range of metrics to assess the quality of synthetic time series datasets.

- We enable conditional time series generation, allowing for conditioning either on a scalar value or temporally.

- We offer a suite of tools for developing time series generative models, including access to a diverse collection of more than 140 popular time series datasets, preprocessing routines, augmentation techniques, and visualization tools.

The code, documentation, and introductory examples are released under the Apache 2.0 license and are available at https://github.com/AlexanderVNikitin/tsgm. TSGM's documentation is available at https://tsgm.readthedocs.io/en/latest/.

## 2 The TSGM framework

The framework we propose supports time series generation for raw data, labeled data (when each time series has a corresponding label), and temporally labeled data (when labels are indexed temporally). It addresses these variations with a unified interface; therefore, the rest of the paper will not differentiate between these types of problems.

Let us start by categorizing the approaches for time series generation into simulation-based and data-driven methods (see Fig. 2). Simulation-based methods generate time series based on a user-defined simulator. One example of a simulation-based method is the hemodynamics simulators used to study cardiovascular systems [75]. Nonetheless, despite the availability of the simulators, generating realistic data requires finding the optimal parameters. By contrast, data-driven methods do not rely on simulators but instead utilize available historical data and a black-box generative model. The majority of previous works on synthetic time series generation have focused on data-driven approaches. TSGM brings these directions together and implements both simulation-based and data-driven methods,

| | Platform | Aug-s | TC | Generators | | | Evaluators | | | | | | |
|---|---|---|---|---|---|---|---|---|---|---|---|---|---|
| | | | | NNs | Prob. | Simul. | Dist. | PC | Priv. | Fairn. | DE | Divers. | Qual. |
| TimeSynth [48] | NumPy | ✗ | ✗ | ✗ | ✓ | ✓ | ✗ | ✗ | ✗ | ✗ | ✗ | ✗ | ✗ |
| DeepEcho [82] | PyTorch | ✗ | ✗ | ✓ | ✓ | ✗ | ✓ | ✗ | ✗ | ✗ | ✗ | ✓ | ✓ |
| SynthCity [59] | PyTorch | ✗ | ✗ | ✓ | ✗ | ✗ | ✓ | ✗ | ✓ | ✓ | ✓ | ✓ | ✓ |
| TSSurrogates [28] | Julia | ✓ | ✗ | ✗ | ✓ | ✗ | ✗ | ✗ | ✗ | ✗ | ✗ | ✗ | ✗ |
| Gretel [3] | PyTorch | ✗ | ✗ | ✓ | ✗ | ✗ | ✓ | ✗ | ✗ | ✗ | ✗ | ✓ | ✓ |
| TSGM | Keras | ✓ | ✓ | ✓ | ✓ | ✓ | ✓ | ✓ | ✓ | ✓ | ✓ | ✓ | ✓ |

Table 1: Comparison of time series generative frameworks. The augmentations column ("Aug-s") compares the availability of classic augmentations algorithms, such as slice and shuffle or magnitude warping. We also compare the available generators (based on NNs, probabilistic, and simulation-based), evaluation approaches (distance, predictive consistency (PC), privacy, fairness, downstream effectiveness (DE), and qualitative comparison; see Sec. 3), and whether the frameworks provide temporally conditioned generation (TC).

allowing for a unified approach to time series generation that bridges the gap between the synthetic data and simulation-based communities.

**Data-driven generators.** Data-driven generators produce synthetic data from a collected historical dataset. These generators have enjoyed significant success in image generation, and we adopted many ideas from computer vision. In TSGM, we implement GANs, VAEs, and probabilistic models for data-driven generation.

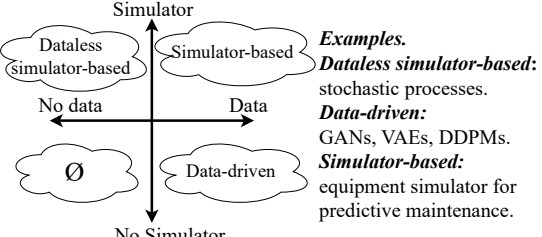

Figure 2: The taxonomy of generative methods in TSGM. Simulation-based methods allow users to specify a simulator. Data-driven methods do not require users to specify the generative process and model time series purely from data.

Generative adversarial networks (GANs, [26]) are a class of machine learning methods where two NNs, a generator and a discriminator, compete in a game where the generator $G$ attempts to generate realistic examples, and the discriminator $D$ strives to distinguish between the generated and real samples. GANs have gained popularity due to their success in generating images and texts, and they have later been applied to time series generation. GANs can be adapted to time series data by choosing suitable generators and discriminator architectures. We provide several architectures in Zoo (architecture repository in TSGM); see Sec. 4 for more detail. The framework allows the training procedures to be extended to Wasserstein GANs, adding a gradient penalty or implementing differentially private approaches. Moreover, we implement models based on the extension of this idea, such as TimeGAN [78], RCGAN [23], and WaveGAN [20]. In addition, TSGM employs conditional and temporally conditional GAN-based models.

Autoencoders are neural network models that consist of two parts: an encoder $E$ and a decoder $D$. The encoder learns to efficiently encode input into a latent representation (the last layer of $E$ is often called a bottleneck), and the decoder learns to recover input using this latent representation. Formally, the generative model can be written as $p(\boldsymbol{x}, \boldsymbol{z}) = p_\theta(\boldsymbol{z})p_\theta(\boldsymbol{x}|\boldsymbol{z})$, where $p_\theta(\boldsymbol{z})$ is a prior Gaussian distribution over latent variables, $p_\theta(\boldsymbol{x}|\boldsymbol{z})$ is the likelihood with parameters computed by the decoder network $D$. The posterior is approximated by $p_\theta(\boldsymbol{z}|\boldsymbol{x}) \approx q_\phi(\boldsymbol{z}|\boldsymbol{x})$, where $q_\phi(\boldsymbol{z}|\boldsymbol{x})$ is usually Gaussian with parameters computed by the encoder network. The Variational Autoencoder (VAE, [35]) model is trained by maximizing the evidence lower bound: $\text{ELBO}_{\phi,\theta} = \mathbb{E}_{q_\phi(\boldsymbol{z}|\boldsymbol{x})}\left[\log p_\theta(\boldsymbol{x} \mid \boldsymbol{z})\right] - D_{\mathbb{KL}}\left(q_\phi(\boldsymbol{z} \mid \boldsymbol{x})\|p_\theta(\boldsymbol{z})\right)$. In beta-VAE [30], the model is trained via constrained optimization, resulting in an adjustable weight between likelihood and KL terms in the objective, which helps to disentangle the representation, forcing the model to produce interpretable latent vector components. In TSGM, we adopt beta-VAE for time series generation. VAE model in TSGM supports many different architectures for encoder and decoder, and in particular, allows users to use TimeVAE [19]. Moreover, TSGM implements conditional and temporally conditional VAE-based models.

**Augmentations.** Often used in computer vision, basic data augmentations enable drastic improvements in deep learning training [9, 76]. Similarly, many augmentations are available in the time series domain. In TSGM, we implement augmentations in the module `tsgm.models.augmentations`,

including noise injection, flipping [60], window-slicing [38], window-warping [60], and dynamic time warping barycenter averaging [24].

**Simulation-based generators** In TSGM, we call simulation-based generators a broad generative model class that includes all user-specified parametric models. Simulation-based method are actively used in numerous scientific fields such as astronomy [17], particle physics [8], econometrics [49], and biology [70]. The training of such models can be performed using standard techniques, such as maximum likelihood estimation or approximate Bayesian computation (ABC, [4, 15]) methods when the likelihood is unavailable. In particular, TSGM implements rejection sampling for those cases. TSGM currently supports stochastic periodic simulators, the Lotka-Volterra model [46], and a predictive maintenance simulator [51]. To determine the parameters of simulators, various methods have been developed, including rejection sampling [67], likelihood-free inference by ratio estimation [69], and neural likelihood estimation [67]. TSGM uses an approximate rejection sampling approach for simulation-based inference and offers a wide range of summary statistics for time series data.

**Model Selection.** Model selection should be performed empirically by running multiple models on a dataset and comparing them with respect to the most relevant metrics for a given application (discussed in Sec. 3). TSGM is apt for this purpose as it permits quickly running a diverse set of models and evaluation metrics. Moreover, TSGM can be used with Optuna [1] for automated models and hyperparameter selection via grid search, Bayesian optimization, and other methods. We provide a tutorial on advanced uses of model selection in our GitHub in the tutorials section.

## 3 TSGM: Evaluation of Synthetic Data

The quality assessment of synthetically generated data depends on the use case. In TSGM, we have designed a range of metrics to cover most such use cases. First, let us describe two primary scenarios of synthetic time series application.

**Scenario 1. Data sharing.** An organization wishes to employ an outside agent to analyze sensitive data. Sharing real data can be complicated due to privacy or commercial concerns. Synthetic counterparts can provide a convenient solution to this problem if the analysis results obtained using synthetic data are transferable to the original data.

**Scenario 2. Data augmentation.** An ML model needs to be trained on a relatively small dataset. However, the dataset is not sufficient for the desired quality of modeling and the model size. Such limited datasets can be augmented with synthetic data. This synthetic data, which must be similar to real data, aims to enhance the model's performance or, in other cases, assist in model reliability tests. Examples of this scenario include, for instance, training time series foundation models [2, 16].

It is crucial to note that while both scenarios call for synthetic data, the quality evaluation of these data depends on the case. Thus, the evaluation methods should take into account the various characteristics of the data. We suggest the following classification for types of evaluation metrics (with the corresponding scenarios in brackets): (i) real data similarity / distance (Sc. 1 and 2), (ii) predictive consistency (Sc. 1), (iii) privacy (Sc. 1), (iv) fairness (Sc. 1 and 2), (v) downstream effectiveness (Sc. 2), (vi) diversity (Sc. 1 and Sc. 2), and (vii) visual comparison (Sc. 1 and 2). Let us discuss each of the evaluation types in more detail. It is important to note that these are *metric types*, and each type has multiple specific implementations in TSGM.

**Similarity / distance.** This metric class evaluates the similarity of the generated and real data. We implement several specific metrics: distance in the space of summary statistics, Maximum Mean Discrepancy (MMD, [27]), and discriminative metric [78]. The summary static distance measures distance in the space of summary statistics $\boldsymbol{S}(D) : \mathrm{D} \to \mathbb{R}^s$, where function $\boldsymbol{S}$ transforms the dataset into a vector of $s$ summary statistics. The distance measure is $\rho(\boldsymbol{S}(D) - \boldsymbol{S}(\widehat{D}))$, where $\rho$ is a norm in $\mathbb{R}^s$. Discriminative metric evaluates a classifier trained to distinguish real and synthetic data.

**Predictive consistency.** This metric assesses the consistency of predictive performance across a set of models on both synthetic and real data. It estimates the likelihood that if one model outperforms another on synthetic data, the same performance relationship will hold on the real data. More formally, for a given set of functions M, $\pi(\mathrm{M}, D, \widehat{D}) = \frac{|\{(m_1, m_2) | m1 \sim m2, m_1 \in M, m_2 \in M\}|}{|M|(|M|-1)}$, where $m_1 \sim m_2$ denotes consistency of predictive performance on real dataset $D$ and its synthetic counterpart $\widehat{D}$ and $|M|$ denotes the cardinality of $M$.

**Privacy.** Privacy of generated data is crucial for many applications. One way to measure this privacy is by evaluating the effectiveness of various attacks on the data. For instance, a common metric is to measure the precision of membership inference attacks, as outlined by Shokri et al. [65]. We propose the following evaluation procedure: 1. Split the historical data into training and hold-out sets ($D_{tr}$ and $D_{test}$), 2. Train a generative model on $D_{tr}$ and generate a synthetic dataset $\widehat{D}$, 3. Train a one-class classification (OCC) model on synthetic data $\widehat{D}$ and evaluate it on $D_{tr}$ and $D_{test}$, 4. Use the precision of the OCC model as the target score.

**Fairness.** Considering the fairness of synthetic data is essential because it can impact the fairness of downstream models. For example, synthetic data can help avoid biases against minorities. TSGM currently uses demographic parity [13] and predictive parity [74] as fairness metrics.

**Downstream effectiveness.** Synthetically generated data can enhance the effectiveness of a downstream task. To evaluate this utility, we compare the performance of a model trained on historical data with a model trained on both historical and synthetic data. In some cases, downstream effectiveness can also be measured using different training/testing allocations. By evaluating several train/test splits, we obtain confidence intervals, resulting in statistically more reliable results.

**Diversity.** The generated data should be sufficiently diverse. The diversity of generated data affects other metrics, such as downstream effectiveness or consistency, but measuring it helps for more detailed specifications. TSGM implements pairwise distance, the Shannon entropy, the spectral entropy [64], and other metrics as a measure of time series diversity.

**Qualitative comparison.** Qualitative evaluation of generated data is as important as quantitative evaluation. Visual exploration can reveal issues such as mode collapse in GANs or critical regimes of the generative model. To facilitate this, we use t-distributed stochastic neighbor embedding (t-SNE, [31]), spectral density visualization, and plain plotting of generated time series, among other methods.

## 4 Framework Design

TSGM provides a unified API that allows users to effortlessly switch between different methods, such as data-driven and simulation-based approaches, to select the most effective one. Additionally, TSGM is extensible, enabling users to implement new methods or extend existing ones easily, allowing for experimentation with custom optimizers, metrics, and architectures. The framework's reliability is ensured through comprehensive testing, static type checking, and experiments. The high-level architecture of the system is illustrated in Fig. 1. The framework is built on Keras because it offers a detailed and flexible API, making it well-suited for creating unified yet customizable models for synthetic time series. Moreover, recent versions of Keras support various backends, including TensorFlow, PyTorch, and JAX, allowing for the extension of TSGM models to these different backends as well.

**Architecture Zoo.** Architecture Zoo is a repository of neural network architectures that framework users can utilize for various purposes, such as solving downstream problems or serving as subcomponents in generative methods, for example, as a discriminator or generator in GANs. Each model has been tested in various experimental settings, making them practical for users. The repository includes CNN-, RNN-, and transformer-based architectures for time series analysis.

**Metrics.** The Metrics module implements the evaluation procedures described in Sec. 3. It is closely integrated with the Statistics module, which introduces summary statistics and evaluates generated data. Additionally, the Metrics module is connected to the Architecture Zoo, using its downstream models to assess the quality of generated data in terms of downstream gain and predictive consistency.

**Optimizers.** This module aims to enhance standard optimization procedures to suit synthetic data generation scenarios. For example, it incorporates an approximate Bayesian computation engine for selecting simulator parameters. Additionally, this module can be used to implement differentially private optimizers for deep learning methods, among other potential applications.

**Performance.** The code can be executed on CPUs, GPUs, and TPUs, and perform distributed training, which significantly improves the performance.

**Differential privacy.** The implemented algorithms can be extended to their differentially private (DP) versions using DP optimizers for training (e.g., GAN models can be transformed to DP-GAN [77]). TSGM is compatible with differentially private optimizers from TensorFlow Privacy.

**Built-in datasets.** TSGM offers convenient access to datasets spanning diverse domains, including samples from Gaussian processes, data sourced from the UCR and UCI repositories [14, 5], and historical stock prices. Currently, TSGM hosts more than 140 datasets (with the majority from the UCR repository [14]) from various fields, such as neuroscience and audio processing, with the collection continuously expanding.

**Reliability and maintenance.** The framework has undergone thorough testing across various operating systems (Windows, Linux, and MacOS) and computational environments (CPU, GPU, TPU, and distributed training). It adheres to software engineering best practices, encompassing unit testing, adherence to PEP-8 styling guidelines, and utilization of static type-checking with MyPy [1]. The comprehensive documentation is available online, along with the guidelines for new developers joining the project. The support of the project is done via Issues on GitHub. Automated testing and style checks are performed via GitHub CI[2].

## 5  Related Work

**Time series generation.** Time series generation methods stem from statistical techniques used for forecasting. In Dunne [22], the authors considered the generation of time series data using AR, ARMA, MA, and ARIMA models. These approaches were effective at simulating the simplest time series. However, these methods fall short in capturing intricate dependencies within time series, making them unsuitable for many applications. Consequently, more sophisticated methods have emerged. Various GAN architectures have been tailored for specific applications such as medical data generation [71], audio generation [53], or financial modeling [78]. Synthetic time series as a proxy for real data have been used, for instance, in predictive maintenance of workstations [51]. In our work, we develop a generic framework for time series synthetic generation problems that connects the listed research directions and provides a platform for new methods by giving interfaces and pipelines. Several works consider improvements to GANs and their effect on the resulting models. For instance, Ni et al. [50] proposed Wasserstein GANs as a method for adding Lipschitz constraints to the space of generated functions. Other methods built on the GAN idea include RCGAN [23], TimeGAN [78], GMMN [33], DoppelGANger [44], WaveGAN [20], and TTS-GAN [42]. VAEs have also been leveraged for synthetic time series generation, exemplified by TimeVAE [19] and TimeVQVAE [41, 40]. Recent methods for synthetic time series generation include diffusion models [66, 58, 80] and transformer-based models [45, 12, 42].

**Tools.** Currently, there are limited software options available for generating synthetic time series data. We outline the advantages of TSGM in Table 1. The key differences lie in offering a more diverse set of metrics, supporting conditional time series generation, and providing a broader range of implemented techniques. For instance, synthetic time series were generated in TimeSynth [48]. However, this tool provides only the most straightforward dataless simulation-based approaches. In contrast, our tool enables data-driven and combined models for time series generation, covering a more significant fraction of use cases. Patki et al. [56] introduced an open-source ecosystem for synthetic data generation in different domains; it implements basic GAN and probabilistic autoregressive models for time series in their framework DeepEcho [82]. Synthcity [59] developed a PyTorch-based framework for broadly defined tabular data and includes some synthetic time series generation methods as well. Compared to this work, the TSGM interface is lower-level (it allows practitioners to experiment with underlying architectures easily, see Sec. 4 for more detail), provides augmentation techniques, temporally conditioned generators, and a broader set of metrics tailored to synthetic time series generation. Haaga and Datseris [28] introduced a Julia library that implements several augmentation methods without focusing on metrics and deep learning approaches. Importantly, TSGM methods can be easily extended to multi-platform usage in TensorFlow, PyTorch, and Jax, thanks to Keras 3.0 multi-platform support.

## 6  Demonstration

In this section, we provide code examples and demo experiments utilizing TSGM. We also offer supplementary examples in tutorials and documentation released in open source.

---

[1] https://mypy-lang.org/        [2] https://resources.github.com/ci-cd/

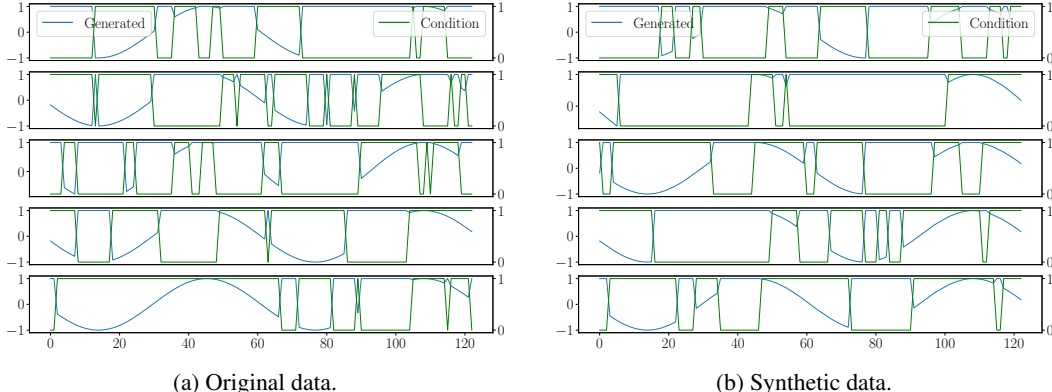

(a) Original data.                    (b) Synthetic data.

Figure 3: Fig. 3a shows the original temporally labeled time series, and Fig. 3b presents synthetic data generated by cGAN. Each graph shows conditions (green lines) and time series (blue lines). We can observe that synthetic data resembles the pattern of the original data.

**Demo 0. TSGM code and API.** We present examples of using our framework in Fig. 1, and additional examples in App. B. The code demonstrates concise and intuitive implementation, making it accessible even without prior knowledge of generative modeling, thus enabling widespread application.

**Demo 1. Selecting a generative model.** In this demonstration, we illustrate how TSGM can facilitate benchmarking and selection of synthetic time series generation models. Specifically, we experiment with three most widely used architectures: TimeVAE, TimeGAN, and RCGAN. In these experiments, we quantified

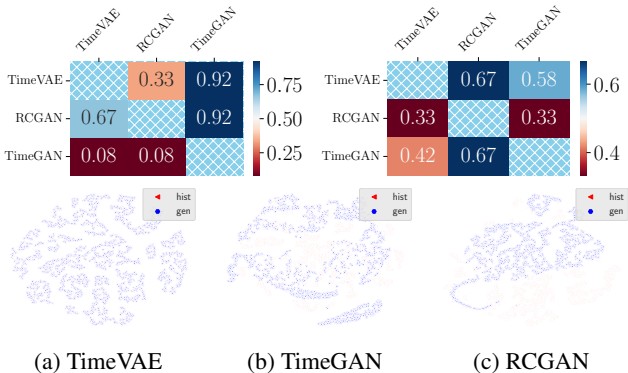

(a) TimeVAE    (b) TimeGAN    (c) RCGAN

Figure 4: Comparing the methods for data sharing (left) and data augmentation (right) tasks across three datasets. The values represent the fraction of cases (metric and dataset pairs) where a method from a row performs better than a method from a column. t-SNE visualizes individual historical (◀) and generated (●) time series.

similarity metrics by computing the Euclidean distance between vectors representing summary statistics extracted from time series data. These vectors were 200-dimensional, particularly in the case of sine waves. We then assessed downstream metrics related to improvement and consistency in autoregression tasks. To evaluate the improvement, we augmented training data with 100 synthetic time series samples and measured MSE improvement on the autoregression task. Consistency was measured by employing three LSTM models with varying layers, ranging from two to four. We reported the percentage of models demonstrating consistency among those LSTMs (with higher percentages indicating better performance). Privacy was evaluated by computing one minus the precision of a membership inference attack using a holdout dataset, resulting in a score between 0 and 1 (with higher values indicating better privacy protection). For the membership inference attack, we utilized a one-class Support Vector Machine (SVM) and concatenated all features for simplicity, although custom attacker models can also be integrated into the framework. All the metrics are implemented and released in TSGM.

We conducted comprehensive experiments utilizing NASA C-MAPSS (turbofan engine degradation simulation dataset), periodic time series data, and UCI energy datasets. Our main goal in these experiments was to showcase the tool's ability to generate realistic synthetic time series data across a range of diverse datasets. To measure the quality of the generated data, we used the metrics defined above. In Fig. 4, we demonstrate the experimental results in the form of summary heatmaps for two primary scenarios outlined in Sec. 3 and individual TSNE plots for each method. For the sharing scenario (shown on the left of Fig. 4), the evaluated metrics are consistency and privacy. In contrast, for the augmentation scenario (on the right of Fig. 4), the metrics include similarity and downstream

effectiveness. It is evident that employing synthetic data for various purposes, such as data sharing and data augmentation, requires different modeling approaches. This highlights the significance of defining a varied set of metrics and emphasizes the need to evaluate synthetic data using TSGM.

**Demo 2. Data augmentation for performance improvement.** We conducted experiments using basic augmentations from TSGM, demonstrating their ability to enhance datasets and improve predictive performance. The experiment used the ECG-200 dataset from Olszewski [54], which comprises of electrical activity during heartbeats. This dataset consists of 100 train and 100 test time series with 96 timesteps. Each time series is labeled with a class indicating whether the record corresponds to a normal heartbeat or myocardial infarction. We compared the effectiveness of augmentations using a 1D-convolutional neural network as a baseline model. This network consists of a 1D-convolutional layer with 64 filters of size 3, followed by dropout with a probability of 0.2, and then a fully connected layer with 128 neurons utilizing ReLU activation. Training the network involved optimizing binary cross-entropy with the Adam optimizer and a learning rate of 0.001. The results are summarized in Table 2, demonstrating that TSGM augmentations can improve the modeling results on the ECG benchmark. We compared the baseline performance with various TSGM augmentations, including Gaussian noise, slicing and shuffling time series, flipping the axis, magnitude warping, and dynamic time warping barycenter averaging. This demonstration underscores the utility of employing easily accessible augmentations from TSGM to improve downstream performance in time series problems.

**Demo 3. Temporally conditioned time series generation.** Next, we demonstrate the effectiveness of conditional data generation within TSGM. Specifically, we assess conditional models on periodic data, where time series are conditioned on a sequence of labels $y_t \in \{0, 1\}$ sampled from the original distribution (for simplicity; in practice, it can be sampled from a separate generative model). The generative process is defined as $x_t = \mathbf{1}[y_t = 1]s\sin(t + C) + \mathbf{1}[y_t = 0]s.$, where $s$ and $C$ are randomly sampled for each feature. We employ a GAN with a recurrent generator and

Table 2: Comparison of augmentations from TSGM on ECG-200 with CNN$_{1d}$ classifier. We use 100 samples to augment the data.

| Augmentation | Accuracy | F1 |
|---|---|---|
| — | $0.89 \pm 0.02$ | $0.89 \pm 0.02$ |
| GaussianNoise | $0.89 \pm 0.01$ | $0.89 \pm 0.01$ |
| SliceAndShuffle [72] | $0.89 \pm 0.01$ | $0.89 \pm 0.01$ |
| Rotation | $0.90 \pm 0.02$ | $0.90 \pm 0.02$ |
| MagnitudeWarp [38] | $0.90 \pm 0.01$ | $0.90 \pm 0.01$ |
| WindowWarp [61] | $\mathbf{0.91 \pm 0.01}$ | $\mathbf{0.91 \pm 0.01}$ |
| DTWBA [24] | $0.90 \pm 0.01$ | $0.90 \pm 0.01$ |

discriminator to produce time series that mimic the pattern and generate diverse, realistic samples. Using visualization utilities from TSGM, Fig. 3a displays the original dataset, while Fig. 3b presents the synthetic data produced by GAN. Notably, we observe a visual resemblance between the original and synthetic data, indicating that the GAN effectively captures the underlying process.

**Demo 4. Benchmarking performance with hardware accelerators.** For performance evaluation, we conducted experiments by running our framework on different hardware configurations: *(i)* CPU training using Intel(R) Xeon(R) CPU @ 2.20GHz processor, *(ii)* GPU

Table 3: Performance comparison.

| Hardware | Two first epochs | Next two epochs |
|---|---|---|
| CPU | 25m 39.33s $\pm$ 2m 1.51s | 26m 49.67s $\pm$ 38.23s |
| GPU | 24.94s $\pm$ 0.02s | 20.48s $\pm$ 0s |
| 2xGPU | 34.36s $\pm$ 0.46s | 19.08s $\pm$ 0.07s |
| TPU | 1m 34s $\pm$ 18.25s | 16.52s $\pm$ 0.63s |

training using NVIDIA Tesla V100, *(iii)* two GPU training using NVIDIA Tesla V100, and *(iv)* TPU training using TPU v2. We trained a cGAN model with 5000 univariate time series conditioned on a binary variable for two epochs (refer to `tsgm.utils` for more details). The results are presented in Table 3. Importantly, running TSGM models on various accelerators does not require additional effort from users.

**Demo 5. Qualitative assessment and monitorings.** TSGM offers a range of methods for visualizing time series data, including side-by-side line plots, t-SNE, and comprehensive summaries of model training. As an illustration of such a training summary, we present the TimeGAN training summary in Fig. C.1. This plot displays the training loss along with t-SNE visualizations of the results available via TSGM monitorings.

**Demo 6. Command Line Interface (CLI).** Alongside the library, we provide a collection of Command-Line Interfaces (CLIs) tailored for users who prefer to avoid engaging directly with programming languages. These CLIs offer a higher-level interface, allowing users to create and evaluate synthetic data directly from the terminal. Currently, the TSGM CLIs include `tsgm-gd`, which generates synthetic data by taking a path to real data and producing synthetic counterparts, and

`tsgm-eval`, which evaluates synthetic dataset quality. An example of how to use `tsgm-gd`is provided in Listing 1.

```
1  tsgm-gd -n-epochs=100 -architecture-type="gan" -latent-dim=8 -source-data=hist
      .pkl -source-data-labels=hist_labels.pkl -dest-data=synth_data.pkl
```

Listing 1: An example of a tsgm-gd run. The CLI generates synthetic data using conditional GAN architecture using serialized real data as input.

**Additional demos.** We provide more demonstrations and tutorials in the TSGM repository, covering topics such as the use of evaluation metrics, augmentations, generators, and datasets.

## 7 Conclusion and Discussion

This paper introduced the *Time Series Generative Modeling framework (TSGM)*, an open-source library for synthetic time series generation. This framework implements a broad range of approaches to synthetic time series dataset generation and evaluation, enabling users to apply both data-driven and simulation-based methods in customized settings. The framework simplifies the development of new datasets and techniques by providing numerous useful abstractions, routines, and built-in datasets for synthetic time series. Additionally, the paper discussed various scenarios for synthetic time series generation and presented the extensive set of metrics we have developed to measure the quality of the generated data. These metrics are implemented within the framework. Moreover, we empirically evaluated several time series generation approaches and provided a selection of ready-to-use architectures that machine learning practitioners can readily employ.

**Impact.** In recent years, numerous methods have been developed for synthetic time series dataset generation. However, these methods have not been as widely adopted as generative techniques for images and texts. This limited usage is due partly to the significant gap between research and practical applications, and partly to the fact that these methods remain at an early stage of development. Our framework addresses both of these issues. Firstly, it allows practitioners to use ready-made methods without the need for a deep understanding of the underlying implementations. Secondly, it provides a platform for researchers to implement and compare new methods, facilitating further development and application of synthetic time series generation techniques.

**Ethics.** Synthetic data generation can significantly impact the modeling process for various problems. It is the responsibility of the machine learning and data mining communities to ensure that these methods are evaluated for fairness and security. Despite their importance, privacy and fairness metrics are often overlooked by researchers in synthetic time series generation. TSGM provides a convenient platform for experimenting with these metrics and incorporating them into the modeling process. Additionally, generative models can be misused to create fake data, such as fake audio recordings. We believe that providing open-source implementations of synthetic data generators would help advance research in detecting fake data by offering examples of synthetic datasets and promoting the open-source development of detection methods. To reduce this risk, robust watermarking algorithms [6] are essential, and we aim to support the development of such algorithms within TSGM.

**Limitations.** Although the codebase is designed to support irregularly sampled time series data, TSGM has not yet been tested with this type of data and currently lacks state-of-the-art models in this domain. We plan to address these gaps in future releases. Additionally, generating longer sequences remains a methodological challenge in time series generative modeling. TSGM provides a platform where new methods can be easily tested on datasets of varying lengths.

## Acknowledgments and Disclosure of Funding

This work was supported by the Research Council of Finland (Flagship programme: Finnish Center for Artificial Intelligence FCAI), UKRI Turing AI World-Leading Researcher Fellowship, EP/W002973/1, and ELISE Networks of Excellence Centres (EU Horizon:2020 grant agreement 951847). We also acknowledge the computational resources provided by the Aalto Science-IT Project from Computer Science IT.

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

# Supplementary Material:
# TSGM: A Flexible Framework for Generative Modeling of Synthetic Time Series

## A  Demonstrations

### A.1  Sine-const dataset

We performed a series of experiments to verify the developed models. In one of the experiments, we generated 5000 temporally labeled time series. We used the following method to produce time series. Target values were randomly alternated between zero and one, with the probability of switching to 0.1. When the target value was equal to one, features were generated using periodic functions with random shifts.

We trained the conditional GAN model using the Adam optimizer with a learning rate of 0.0001. An exponential decay rate for the first moment estimates equals 0.5; as a loss function, we use binary cross-entropy. Discriminator architecture consisted of four one-dimensional convolutional layers with a leaky-ReLU activation function, a global average pooling layer, and a fully connected layer with a sigmoid activation function. Generator architecture consisted of three one-dimensional convolutional layers with leaky ReLU activations and dropouts with a rate equal to 0.2, followed by an LSTM layer with the same dropout, and then by average pooling and locally connected layer. We have trained the model for 1000 epochs; however, it converged earlier. The confidence intervals were obtained by bootstrapping 90% of the synthetic data.

### A.2  NASA C-MAPPS (Demo 1)

NASA C-MAPPS dataset contains measurements from the simulation of realistic large commercial turbofan engine data. This dataset is often used to model the degradation of those engines, for instance, for predictive maintenance applications. Each simulation is described by 28 sensors, but for these experiments, we manually selected 12 sensors correlated with the engines' conditions. We split the dataset into shorter time series of length 24 and used 1100 time series for training.

### A.3  UCI Energy dataset (Demo 1)

We also evaluated some of the implemented methods on the appliances energy dataset, which consists of 4.5 months of 10-minute measurements. The measurements include temperature, pressure, and data from various sensors inside the building (in total 28 attributes). We split the dataset into shorter time series of 16, and we used 1000 time series for training and 233 for holdout.

### A.4  Datasets

TSGM allows for a unified loading of various popular time series datasets such as UCR [14], Mauna Loa $CO_2$ (CC BY 4.0 Deed, [34]), EEG eye state (CC BY 4.0, [62]), Individual Household Electric Power Consumption (CC BY 4.0, [29]), COVID-19 over the US (CC BY-NC, [52, 68]), Appliances Energy Prediction (CC BY 4.0, [11]), MNIST (Yann LeCun and Corinna Cortes hold the copyright of MNIST dataset, which is a derivative work from original NIST datasets. MNIST dataset is made available under the terms of the Creative Commons Attribution-Share Alike 3.0 license, [39]), samples from a Gaussian process, and Physionet 2012 Challenge (Open Data Commons Attribution License v1.0, [57]).

Also, we asked our industrial partner for feedback on how long it takes to (a) use the provided library in a production pipeline and (b) use one of the pre-existing methods with an open-source implementation. They estimated that (a) takes from a couple of days to one week, while (b) takes one week to one month time (and this is just for one method — incorporating and comparing several methods would sum up because, often, it is not much shared between them. Even more time would be saved in cases without open-source models and for evaluation procedure implementation. These estimates align with our experience working with other open-source implementations of time series generation models.

# B Additional Examples

Here, we provide additional examples of various framework applications.

The framework is concise and comprehensible, as can be seen in a minimal example of data-driven synthetic time series generation provided in Listing 2.

```python
1  from tsgm.models.cgan import GAN
2  from tsgm.models import zoo
3  # ... necessary optimizers and losses
4  # ... define original data and hyperparams
5
6  # real_data: shape n, seq_len, n_features,
7  arch = zoo["cgan_base_c4_l1"](
8      seq_len=seq_len,
9      feat_dim=feature_dim,
10     latent_dim=latent_dim,
11     num_classes=output_dim)
12 gan = GAN(
13     discriminator=arch.discriminator,
14     generator=arch.generator,
15     latent_dim=latent_dim
16 )
17 gan.compile()
18 gan.fit(dataset, epochs=100)
19 generated_samples = gan.generate(100)
```

Listing 2: Synthetics data generation using GANs.

```python
1  import tsgm
2  from tsgm.metrics import PrivacyMembershipInferenceMetric
3
4  # ... Define UserDefinedAttackerModel and train and synthetic
       datasets
5  attacker = UserDefinedAttackerModel()
6  privacy_metric = PrivacyMembershipInferenceMetric(
7          attacker=attacker
8      )
9  priv_res = privacy_metric(
10     tsgm.dataset.Dataset(X_train, y=None),\
11     tsgm.dataset.Dataset(X_syn, y=None),\
12     tsgm.dataset.Dataset(X_holdout, y=None))
```

Listing 3: Privacy evaluation.

In Listing 3, we show how to evaluate the privacy of a generated dataset using TSGM.

In another example (Listing 4), we show an example of downstream gain metric usage. Other metrics are used similarly within the framework.

```
1  from tsgm.models import zoo
2  from tsgm.metrics import DownstreamPerformanceMetric
3
4  # define d_real, d_syn, d_test, and CustomEvaluator
5
6  downstream_model = zoo["clf_cl_n"](
7    seq_len, feat_dim, n_classes, n_conv_lstm_blocks=1).model
8  downstream_model.compile(
9    loss='binary_crossentropy', optimizer='adam',
10   metrics=['accuracy'])
11
12 # CustomEvaluator is a user's class for evaluation
13 evaluator = CustomEvaluator(downstream_model)
14
15 downstream_perf_metric = DownstreamPerformanceMetric(evaluator)
16 print(
17   downstream_perf_metric(d_real, d_syn, d_test))
```

Listing 4: Example of downstream performance metric calculation. CustomEvaluator implements an evaluation strategy to measure the performance gain of a downstream model when synthetic data are used.

```
1  from tsgm.dataset import DatasetProperties
2  from tsgm.simulator import SineConstSimulator
3  from tsgm.optimization.abc import RejectionSampler
4
5  data = DatasetProperties(N=100, D=2, T=100)
6  simulator = SineConstSimulator(
7      data=data, max_scale=max_scale, max_const=20)
8  priors = {
9      "max_scale": tfp.distributions.Uniform(9, 11),
10     "max_const": tfp.distributions.Uniform(19, 21)
11 }
12 samples_ref = simulator.generate(10)
13
14 discrepancy = lambda x, y: np.linalg.norm(x - y)
15 sampler = RejectionSampler(
16     data=samples_ref, simulator=simulator,
17     statistics=statistics, discrepancy=discrepancy,
18     epsilon=0.5, priors=priors)
19
20 sampled_params = sampler.sample_parameters(10)
```

Listing 5: Simulation-based approach to synthetic data generation via ABC. Using rejection sampling the code estimates simulator parameters.

In Listing 5, we show how approximate Bayesian computation is used to estimate the posterior of a model's parameters for simulation-based models.

## C  TimeGAN Training Summary

The implementation of TimeGAN follows the original proposal in Yoon et al. [78]. It consists of an autoencoder component, including an embedding network and a recovery network, and an adversarial component, including a generator and a discriminator. In order to produce the embeddings, the autoencoder component is trained first by minimizing the reconstruction loss (mean squared error, see trace autoencoder loss in Fig. C.1a). When the GAN is trained on embeddings of synthetic data (its own previous outputs), the unsupervised generator loss is minimized (mean squared error, see trace adversarial supervised loss in Fig. C.1a). On the other hand, when the generator is trained on embeddings of real data, the supervised generator loss is minimized. In practice, the network is jointly trained to minimize the generator loss, which is a sum of the first two moments loss, binary cross-entropy loss, and mean squared error loss (see traces generator moments loss, unsupervised generator

loss, and supervised generator loss in Fig. C.1a), At the same time, the joint network training loop also minimizes the embedding network loss (mean squared error, see trace embedder in Fig. C.1a) and the discriminator loss (binary cross-entropy loss, see trace discriminator loss in Fig. C.1a).

The training data utilized for TimeGAN experiments are obtained as in Yoon et al. [78], by generating 6000 multivariate signals of length 24. Each multivariate signal is composed of six sinusoidal features with independently and uniformly sampled values of frequency and phase. T-SNE plots (Fig. C.1b-C.1c-C.1d-C.1e) are produced at various checkpoints during the training loop to compare the quality of synthetic data, in the same way as proposed in Yoon et al. [78] — with averaged feature vectors. On the one hand, t-SNE analysis shows that the results in Yoon et al. [78] were successfully reproduced. Training data (red triangles) and synthetic data (blue hexagons) appear to be almost perfectly mixed at 12000 epochs (Fig. C.1e). On the other hand, it is clear that, even though the network losses (Fig. C.1a) mostly converge after 2000 epochs, the quality of synthetic data continues to improve with the number of epochs.

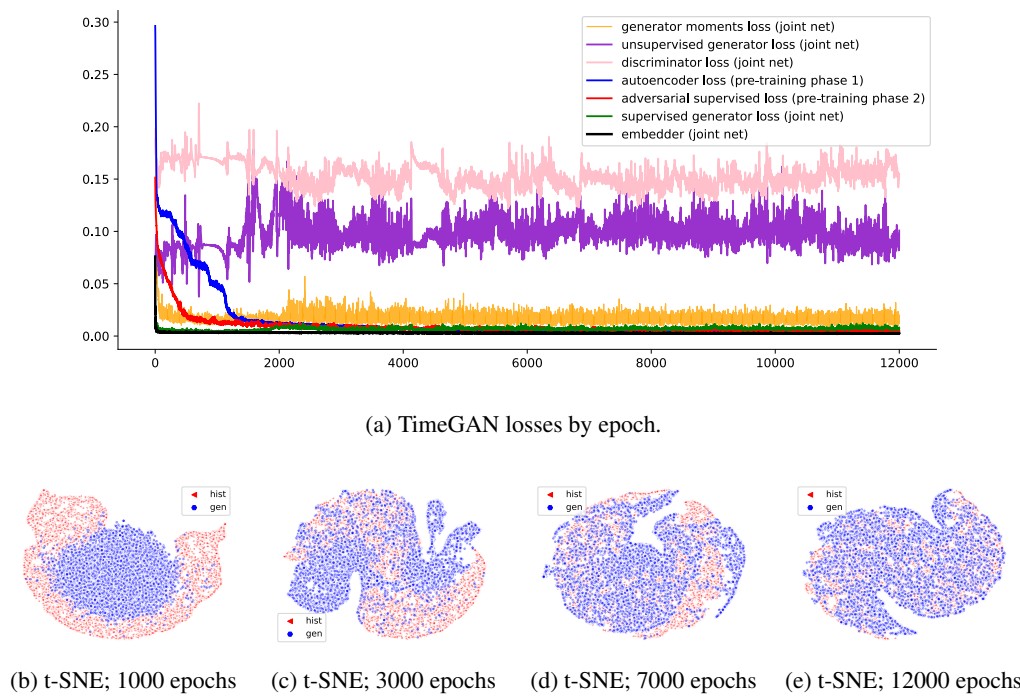

(a) TimeGAN losses by epoch.

(b) t-SNE; 1000 epochs     (c) t-SNE; 3000 epochs     (d) t-SNE; 7000 epochs     (e) t-SNE; 12000 epochs

Figure C.1: TSGM Monitoring. Fig. C.1a shows the values of various losses utilized for training TimeGAN, and Fig. C.1b-C.1c-C.1d-C.1e show t-SNE analysis of training data and synthetic data generated by TimeGAN after 1000 epochs, 3000 epochs, 7000 epochs, and 12000 epochs respectively. Training data are marked with red triangles, whereas synthetic data are marked with blue hexagons.

# D   Metrics

## D.1   Privacy Metric

As we mention in Sec. 3, the privacy of synthetic data is crucial for many applications. The definition of privacy may vary depending on the context. For example, in time series data, each sample can represent an individual, making it essential to prevent information about the original subjects from being leaked into the synthetic data. Specifically, TSGM provides a metric to evaluate the precision of membership inference attacks (see Fig. C.2b) using the following algorithm: 1. Split the historical data into training and hold-out sets ($D_{tr}$ and $D_{test}$), 2. Train a generative model on $D_{tr}$ and generate a synthetic dataset $\widehat{D}$, 3. Train a one-class classification (OCC) model on synthetic data $\widehat{D}$ and evaluate it on $D_{tr}$ and $D_{test}$, 4. Use the precision of the OCC model as the target score.

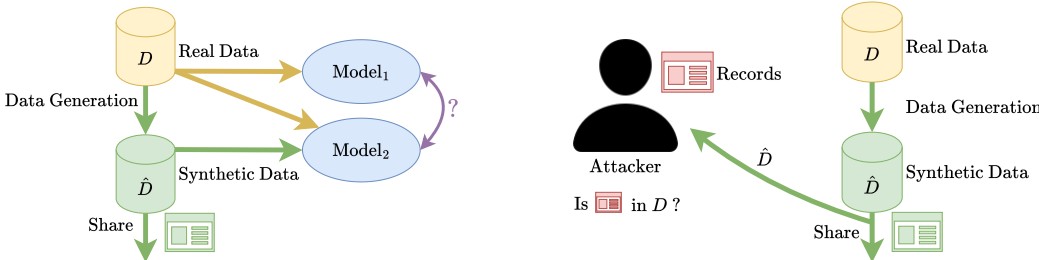

(a) The downstream effectiveness metric compares the performance of Model$_1$ and Model$_2$ on a specific downstream task.

(b) The membership inference attack metric evaluates the precision of attacks aimed at inferring membership, based on the generated synthetic data.

Figure C.2: A schematic illustration of the downstream effectiveness and membership inference attack metrics.

**Example.** One application of this algorithm is generating synthetic trajectories, where a generative model creates time series of coordinates representing individuals' movements. In this context, an attacker may attempt to identify whether a specific individual was part of the training dataset using a shared synthetic dataset. Therefore, it is essential to evaluate the data's robustness against membership inference attacks before sharing it.

## D.2 Downstream Effectiveness

As discussed in Sec. 3, downstream performance can be measured using different train/test allocations. For instance, when the dataset is limited and non-private, it is useful to evaluate downstream gain when a model is trained on the combination of synthetic and real data. However, if the dataset is private, it is more appropriate to train the model exclusively on synthetic data, as the real historical data cannot be shared. A visualization of this can be found in Fig. C.2a.

