# OpenReview forum: "TSGM: A Flexible Framework for Generative Modeling of Synthetic Time Series"
_NeurIPS.cc/2024/Datasets_and_Benchmarks_Track — NeurIPS 2024 Track Datasets and Benchmarks Poster_

### Official Review · Reviewer_S8WW · 2024-07-11
**A Good Foundation for Generative Synthetic Time Series Framework**

**Rating:** 7
**Confidence:** 3
**Clarity:** The paper is well-written and easily …

**Review:**

The paper is well written and addresses a good research topic on Time Series Generative Modeling , which is not yet well-studied in time series area.  This engineering framework is reasonable and it is a good start to build the basic framework for future research work.

**Strengths:**

The paper introduced Time Series Generative Modeling (TSGM), an open-source framework for generating synthetic time series data, which can address the limitations of temporal data availability in machine learning. TSGM encompasses a wide array of machine learning methods, including generative models, probabilistic approaches, simulation-based techniques, and data augmentation strategies. The framework allows users to assess the quality of the generated data from various perspectives such as similarity, downstream effectiveness, predictive consistency, diversity, fairness, and privacy, while being extensible and user-friendly, enabling researchers to quickly implement and compare their own methods in a collaborative environment.

**Additional Feedback:**

n/a

**Correctness:**

There is not much theoretical discussion and the metrics used are quite standard in time series analysis, which seem correct.

**Documentation:**

Looks ok to reproduce.

**Ethics:**

I don't see any ethics problem.

**Limitations:**

It would be nice to see datasets of more statistical properties to be generated and compared, e.g., how to simulate stock prices with regime change.
It would be also better to consider more datasets of longer-terms to understand the differences between short term data generation and long term data generation methods, as well as the metrics.  Maybe the engineering framework would be different too.

**Opportunities For Improvement:**

GANs and VAE-based methods are not the complete set of generation possibilities.   It would be nice to show some more ‘looking-forward’ kind of approaches.

**Relation To Prior Work:**

The paper included the benefits of the previous open source for generative frameworks and provides conditioned generation.

**Summary And Contributions:**

The paper introduced Time Series Generative Modeling (TSGM), an open-source framework for generating synthetic time series data, which can address the limitations of temporal data availability in machine learning. TSGM encompasses a wide array of machine learning methods, including generative models, probabilistic approaches, simulation-based techniques, and data augmentation strategies. The framework allows users to assess the quality of the generated data from various perspectives such as similarity, downstream effectiveness, predictive consistency, diversity, fairness, and privacy, while being extensible and user-friendly, enabling researchers to quickly implement and compare their own methods in a collaborative environment.

---

> ### Author Rebuttal · Authors · 2024-08-17
>
> Dear reviewer S8WW,
>
> thank you for the positive evaluation of significance, usefulness, and evaluation metrics. We would like to address your questions and concerns:
>
> > It would be nice to see datasets of more statistical properties to be generated and compared, e.g., how to simulate stock prices with regime change.
>
> In Demonstrations 1 and 2, we use rather complex datasets, e.g., NASA C-MAPPS and energy UCI dataset. We also added the MIT-BIH Arrhythmia dataset ([PR#54](https://github.com/AlexanderVNikitin/tsgm/pull/54)) to TSGM to facilitate experiments with datasets with regime changes.
>
> > It would be also better to consider more datasets of longer-terms to understand the differences between short term data generation and long term data generation methods, as well as the metrics. Maybe the engineering framework would be different too.
>
> Thank you for pointing this out! We want to emphasize that TSGM is not limited to short-time series data. It can effectively handle longer time series data, allowing researchers to experiment and evaluate synthetic time series generation across different time horizons by simply adjusting a single parameter. However, in the synthetic time series generation community, it is common to use rather short time series slices to evaluate the methods. More work is required on longer time series generation, and tsgm can provide a platform for these experiments. We will add a comment about it in the next revision.
>
> > GANs and VAE-based methods are not the complete set of generation possibilities. It would be nice to show some more ‘looking-forward’ kind of approaches.
>
> Thank you for pointing this out! Now, we have added a DDPM implementation ([PR#56](https://github.com/AlexanderVNikitin/tsgm/pull/56)), and added issues to support normalizing flow ([Issue#55](https://github.com/AlexanderVNikitin/tsgm/issues/55)), Diffusion-TS ([Issue#57](https://github.com/AlexanderVNikitin/tsgm/issues/57)), and TTS-GAN ([Issue#58](https://github.com/AlexanderVNikitin/tsgm/issues/58)).
>
> **Concluding remarks**
> We would be grateful if you could let us know whether our explanations have addressed your concerns. Please let us know if you have any other questions or concerns.

---

> > ### Comment · Reviewer_S8WW · 2024-08-20
> >
> > My concerns have been addressed and I am willing to change my rating to 7.

---

### Official Review · Reviewer_6muW · 2024-07-16
**Reviews**

**Rating:** 7
**Confidence:** 3

**Review:**

#### Pros:

- **Quality**: The TSGM framework is well-implemented, with a robust architecture that supports a variety of machine learning methods for time series data generation.
- **Clarity**: The paper is clearly written, with a logical structure that is easy to follow. The code is well-documented, facilitating user comprehension and adoption.
- **Originality**: The framework introduces novel features such as conditional generation and a comprehensive set of metrics for evaluating synthetic data, which are unique in the current landscape of time series generation tools.
- **Significance**: The work addresses a critical need for synthetic data in applications where privacy and data scarcity are concerns, making it highly relevant to the machine learning community.

#### Cons:

- **User Experience**: While the paper claims the framework is user-friendly, the lack of direct user feedback or case studies leaves some uncertainty about the ease of use for practitioners without a deep technical background.
- **Scalability**: The paper could provide more insight into how the framework performs with large-scale datasets and its efficiency in terms of computational resources.
- **Diversity of Data**: The framework's capability to handle a wide range of time series data types is a strength, but more demonstration on datasets beyond the ones presented would be beneficial to showcase its versatility.
- **Comparative Analysis**: There is limited discussion on how TSGM compares to existing tools in terms of performance, features, and usability, which would be valuable for potential users to make informed decisions.

Overall, the TSGM framework presented in this paper is a valuable contribution to the field of synthetic time series generation. It offers a flexible and comprehensive approach that is well-documented and ethically considered. However, there is room for improvement in terms of showcasing the framework's scalability, user experience, and comparative analysis with existing tools.

**Strengths:**

- **Significance**: The framework addresses a crucial gap in the availability of synthetic time series data for ML applications.
- **Relevance**: It is highly relevant to researchers and practitioners in time series analysis, data privacy, and ML.
- **Quality**: The paper is well-structured, and the code is well-documented and publicly available.
- **Ethical Consideration**: The authors discuss the importance of privacy in synthetic data generation.

**Additional Feedback:**

- In comparison to the qualitative description of the various time-series data generation frameworks presented in Table 1, it would be beneficial for the authors to provide more detailed experimental data. This could include information on whether some of the synthetic data produced by a given method is more beneficial for downstream tasks than other methods.
- The authors might consider adding more information on how TSGM can be integrated into existing ML pipelines.

**Clarity:**

The paper is well-written, but the introduction could be strengthened by providing a clearer problem statement upfront.

**Correctness:**

The claims made in the submission appear to be correct, and the construction of the framework and datasets seems sound.

**Documentation:**

The documentation for the framework is comprehensive, but the paper could include more details on the maintenance and future development plans for TSGM.

**Limitations:**

- The authors acknowledge that TSGM has not been tested with irregularly sampled time series data and does not include state-of-the-art models for this domain.
- The discussion on the potential misuse of generative models to create fake data is appreciated, but concrete steps or features within TSGM to mitigate this risk could be added.

**Opportunities For Improvement:**

- With regard to a number of the indicators proposed in the manuscript, this paper is deficient in terms of the practical application of these indicators and the corresponding experimental results. The paper could benefit from more detailed examples or case studies demonstrating the framework's application in real-world scenarios.
- There is a need for more comprehensive benchmarking against existing tools to highlight TSGM's advantages. It would be beneficial for the authors to explicitly delineate the advantages of their proposed TSGM in comparison to existing frameworks, beyond merely discussing its applicability and uniformity. It would also be advantageous to present comparative results on the tasks, in order to facilitate a more nuanced understanding of the differences and improvements between their work and previous research.

**Relation To Prior Work:**

The paper could improve on this by more explicitly contrasting TSGM with existing tools and frameworks in the field.

**Summary And Contributions:**

This paper introduces the Time Series Generative Modeling (TSGM) framework, an open-source tool aimed at synthetic time series data generation. The framework supports both data-driven and simulation-based methods, offering a unified approach with an extensive set of metrics for evaluating the quality of synthetic data. Key contributions include the development of conditional generation capabilities, a suite of tools for model development, and the provision of over 140 time series datasets for testing and use.

---

> ### Author Rebuttal · Authors · 2024-08-17
>
> Dear reviewer 6muW,
>
> Thank you for positively evaluating the quality, clarity, originality, and significance of our work. We’d like to address pointed weaknesses, and address the remaining questions.
>
> > While the paper claims the framework is user-friendly, the lack of direct user feedback or case studies leaves some uncertainty about the ease of use for practitioners without a deep technical background.
>
> We provide feedback from industrial collaboration in Appendix A5 and briefly discuss anticipated time savings. For users without technical background, we demonstrate tsgm CLI tools that can be used without any coding experience.
>
> > The paper could provide more insight into how the framework performs with large-scale datasets and its efficiency in terms of computational resources.
>
> Importantly, our framework does not introduce additional overhead compared to existing methods, so it should scale as effectively as the original approaches. Additionally, thanks to Keras support, the framework can seamlessly leverage a full range of accelerators and distributed training utilities. We demonstrate the performance analysis using multiple GPUs or TPUs in Tab. 3.
>
> > The framework's capability to handle a wide range of time series data types is a strength, but more demonstration on datasets beyond the ones presented would be beneficial to showcase its versatility.
>
> In the paper, we evaluate the performance of augmentations / compare generations on datasets of various difficulty: periodic, NASA-CMAPPS, ECG, and energy data. For all these datasets, we present some metrics (e.g., show gain achieved by augmentations in Tab. 2, or compare generative methods for different scenarios in Fig. 4). Those datasets are often not that interpretable and we demonstrate TSNE instead of showing the actual signals. We will aim to provide additional examples through the library tutorials.
>
> > there is limited discussion on how TSGM compares to existing tools in terms of performance, features, and usability, which would be valuable for potential users to make informed decisions.
>
> In terms of features, we discuss various frameworks in Table 1, highlighting the key features that differentiate our framework from others. Regarding performance, our framework does not introduce any overhead compared to Keras, which can utilize different backends, such as TensorFlow, PyTorch, and Jax, and typically performs similarly to those models. In terms of usability, we provide feedback from industrial collaboration in Appendix A5, where we briefly discuss the anticipated time savings. For users without a technical background, we offer a CLI tool that can be used without any coding experience.
>
> > With regard to a number of the indicators proposed in the manuscript, this paper is deficient in terms of the practical application of these indicators and the corresponding experimental results. The paper could benefit from more detailed examples or case studies demonstrating the framework's application in real-world scenarios.
>
> Thank you for pointing this out! We present case studies on sharing, and augmenting data in Demonstrations 1 and 2. We will aim to provide more examples through tutorials, available as a part of the framework.
>
> >The discussion on the potential misuse of generative models to create fake data is appreciated, but concrete steps or features within TSGM to mitigate this risk could be added.
>
> Thanks for pointing this out! We believe that adding open-source implementations of synthetic data generators would advance research in detecting fake data by providing examples of synthetic datasets and encouraging the open-source development of detection methods. We will include a comment on this.
>
> **Concluding remarks**
> We would be grateful if you could let us know whether our explanations have addressed your concerns. Please let us know if you have any other questions or concerns.

---

> > ### Comment · Reviewer_6muW · 2024-08-19
> >
> > The authors have addressed my concerns.  I will keep the rating.

---

### Official Review · Reviewer_BaRA · 2024-07-23
**review on the paper TSGM**

**Rating:** 7
**Confidence:** 4

**Review:**

The TSGM library addresses the challenge of acquiring large quantities of time series data (TS) dedicated to scalable training deep-learning models, including the foundational models for time series, receiving significant attention recently. It proposes a generative approach in which substantial time series datasets can be generated from even tiny available datasets on which then a downstream task of choice can be performed. TSGM brings the most comprehensive framework for TS generation and evaluation compared to related work. The paper is generally well-written and easy to follow; the library's main features are discussed in the main paper, while examples of library usage are presented in the paper's Appendix.

**Strengths:**

There is an increasing need for time-series datasets in machine learning dedicated to training deep learning models. Compared to the NLP domain, there is much less curated time-series open data available, which entails risks of overfitting new models to some specific benchmark suite. The work is an important contribution toward extending the available datasets. The comparison of the paper's contribution against the available related libraries demonstrates that it is the most comprehensive method in terms of TS generation and evaluation capabilities.

**Additional Feedback:**

as above

**Clarity:**

The paper is generally well written, additionally, the appendix contains necessary details on using the library in practice with clear usage examples.

**Correctness:**

The library is thoroughly documented and available in open-source providing opportunity for independent evaluation and raising confidence that it is sound.

**Documentation:**

The TSGM library is thoroughly documented and the open-source software is available through a GitHub repository.

**Limitations:**

The authors have adequately addressed the limitations and potential negative societal impact.

**Opportunities For Improvement:**

*  I think the paper would benefit from presenting some down-stream tasks for which the generated datasets can be applied;
* Sharing pre-computed datasets using the presented library with example time series would make it easier to use and apply the contribution in practice;
* Design experiments to answer the question: If the data is generated using a deep GAN / VAE, how can it be avoided that the data be biased due to network properties that could potentially generate out-of-distribution data?

**Relation To Prior Work:**

The paper contains a related work section, and the contribution is compared against the related work in Table 1 along several dimensions.

**Summary And Contributions:**

The paper presents a comprehensive open-source framework for time series generative modeling and evaluation (TSGM). It discusses the structure, implementation, and main features of the TSGM library. It explains how synthetic data can be generated using various generative techniques, such as data augmentation,GAN and VAE trained generative models, probabilistic, and simulation-based, to evaluate the quality of the resulting time series from different angles.

---

> ### Author Rebuttal · Authors · 2024-08-17
>
> Dear reviewer BaRA,
>
> Thank you for positively evaluating the significance, and the usefulness of the proposed framework. We’d like to address your questions and concerns.
>
> > I think the paper would benefit from presenting some down-stream tasks for which the generated datasets can be applied;
>
> In our demonstrations (Fig. 4 and Tab. 2), we use generated data for downstream tasks to illustrate the utility of synthetic data in augmenting the original dataset. We aim to provide more examples in the tutorials available online.
>
> > Sharing pre-computed datasets using the presented library with example time series would make it easier to use and apply the contribution in practice;
>
> We totally agree with this! We are aiming to develop a repository of ready-to-use synthetic datasets, but we are currently unsure with the legal aspect. We’ve created an issue ([Issue#53](https://github.com/AlexanderVNikitin/tsgm/issues/53)) in order to explicitly add to the roadmap.
>
> > Design experiments to answer the question: If the data is generated using a deep GAN / VAE, how can it be avoided that the data be biased due to network properties that could potentially generate out-of-distribution data?
>
> Bias in generated data can be categorized into two main aspects:
> 1) Fairness Bias: To address fairness concerns, we have implemented several fairness metrics, including `metrics.DemographicParityMetric` and `metrics.PredictiveParityMetric`. Evaluating generated data for fairness bias is essential and should be tailored to the specific application.
> 2) Out-of-Distribution (OOD) Bias: To detect discrepancies related to out-of-distribution data and other forms of bias, we use metrics such as `DescriminatorMetric` and `MMDMetric`.
>
> In summary, our framework provides tools for detecting various biases using post-hoc evaluation metrics. Users can then assess whether the generated data is suitable for downstream applications.
>
> **Concluding remarks**
> We would be grateful if you could let us know whether our explanations have addressed your concerns. Please let us know if you have any other questions or concerns.

---

### Official Review · Reviewer_dX7D · 2024-07-25
**A framework for time-series generative models**

**Rating:** 7
**Confidence:** 4
**Correctness:** The submission is correct.
**Clarity:** The paper is clear enough.

**Review:**

Clarity:
- Pro: The paper is generally easy to read, with clear and motivated contributions.
- Cons: However, some areas could benefit from improvements. For instance, Section 2 could offer a more balanced discussion between data-driven generators and simulation-based methods. The discussion on Section 2 is not completely aligned with Figure 2, which mentions also DDPMs (is it a typo?). Additionally, Table 1 is referenced only at the end of the paper, and several acronyms (e.g., PC) become clear only later in the text, making it more time-consuming to search for the best framework and understand why one might select TSGM over another. Some metrics are not discussed enough while they could be further explained on the supplementary material. In particular, those not well-established in the deep learning time-series community, such as the proposed privacy metric. In the limitations the authors mentioned they did not test irregular sampled time-series, but they also include physionet-2012 as dataset.

Quality:
- Pro: The paper and the framework represent a solid contribution. The code available on GitHub appears well-written, commented, and user-friendly, making it straightforward to install and reproduce some tutorials on Colab. The wide availability of datasets is a strong point, though it should be noted that 128 datasets come from the same source.


Originality:
- Pro: The framework is relatively novel, focusing specifically on time-series generative models for augmentation and generation. This niche is not fully addressed by existing work, as highlighted in Table 1. The existing frameworks mentioned are more general and not exclusively tailored to time-series data, making them more complex to use and extend.
- Minor: It is worth noting that Table 1 might contain a typo, as Synthcity, to the best of my knowledge, also supports qualitative metrics like t-SNE visualization. I also suggest to mention SDV in Table 1 for a more comprehensive comparison.


Significance:
- Pro: As mentioned, the work is significant for a niche group of practitioners focusing on time-series data, as the framework is well-focused on this area and is easy to use. It offers out-of-the-box models and numerous datasets, making it accessible and useful. With community support, it has the potential to become a valued tool.

- Cons: On the other hand, the framework may not be immediately relevant to the broader research community on time-series generative models for two reasons. 1) First, the choice of Keras is not adequately discussed or justified; it is unclear whether it is faster or slower compared to other libraries, especially since most recent models are available in PyTorch. 2) Second, the framework does not yet support more recent solutions, such as diffusion models or normalizing flows for time-series. While they could be integrated, the use of Keras might slow this integration and its adoption by the research community. In fact, these solutions are mostly developed in PyTorch.

**Strengths:**

- The work is significant for practitioners working on time-series data, making state-of-art metrics and methods available.
- The framework supports around 140 datasets, which could improve the design and comparison of novel models
- The code is already available on GitHub, and it appears well-written, well commented, and easy to install and run.

**Additional Feedback:**

None

**Documentation:**

There is enough documentation.

**Limitations:**

See discussion above.

Major limitations:

- The choice of Keras is somewhat controversial. It is not adequately discussed or motivated, and most recent time-series models are now in PyTorch.

- More recent time-series generative models, such as diffusion models and normalizing flows, are not considered or discussed.

- Some details are missing, and discussion on simulation or probabilistic models is not detailed enough.

Overall, while I am confident that the framework has merit, it may seem better suited for practitioners and industry rather than the research community.

**Opportunities For Improvement:**

Please see the all comments above.

But in general I suggest to:
- integrate novel models (e.g,. diffusion models and normalizing flow) and discuss relevant papers in the related work, which now stops at VAEs.
- improve writing, motivation, and discussion (e.g., why Keras).
- introduce the predictive score metric [1] : how useful the synthetic data are for a downstream task in the generation case.
- discuss better the supported models, to give a bit more space (maybe appendix) to simulation, and how to find their parameters.

**Relation To Prior Work:**

The authors mentioned previous work, and the main differences are clear enough .

**Summary And Contributions:**

The core contribution of this paper is the introduction of TSGM, a comprehensive framework for time-series generative models. TSGM offers a variety of models, both data-driven and simulation-based, and supports approximately 140 datasets. It also includes a robust set of evaluation metrics specifically designed for time-series data.

Implemented in Keras, TSGM facilitates both augmentation and generation tasks, and it incorporates neural network models like GANs and VAEs. Each task is supported by relevant metrics, enabling the user to find the most suitable model for a given task. Additionally, TSGM is compatible with GPUs and TPUs.

The framework is publicly accessible on GitHub, along with several tutorials to guide users. The authors have also demonstrated the performance of the models across various datasets, showcasing the framework effectiveness.

---

> ### Author Rebuttal · Authors · 2024-08-17
>
> Dear reviewer dX7D,
>
> Thank you for many useful suggestions, and highly evaluating many aspects of our work, including clarity, significance, and the quality of the TSGM codebase. We’d like to address the pointed out disadvantages, and answer questions:
>
> > Section 2 could offer a more balanced discussion between data-driven generators and simulation-based methods
>
> Thank you for pointing this out! To enhance clarity, we will provide a more detailed discussion of simulation-based methods, especially regarding their applications in scientific fields such as astronomy [1], particle physics [2], econometrics [3], and biology [4]. In our work on TSGM, we primarily focus on data-driven methods, reflecting the current trend in the synthetic time series community, which also emphasizes these approaches.
>
> >The discussion on Section 2 is not completely aligned with Figure 2, which mentions also DDPMs (is it a typo?).
>
> Thank you for pointing this out! We will fix Fig. 2 to reflect the state of TSGM in the next revision.
>
> > Tab. 1 is referenced only at the end of the paper, and several acronyms (e.g., PC) become clear only later
>
> Thank you for pointing that out! We will provide more explanations for the acronyms and metrics in the caption and reference the table in the introduction.
>
> > Some metrics are not discussed enough while they could be further explained on the supplementary material [...]
>
> We will include a more comprehensive discussion about the metrics in the Appendix and add additional explanatory visualizations.
>
> > In the limitations the authors mentioned they did not test irregular sampled time-series, but they also include physionet-2012 as dataset.
>
> The current methods are designed for regularly sampled time series data. Although the Physionet-2012 dataset requires preprocessing to be compatible with currently implemented TSGM generators, we also offer the option to access this dataset without preprocessing.
> > The wide availability of datasets is a strong point, though it should be noted that 128 datasets come from the same source
>
> We will add a comment about it in the next revision.
>
> > Table 1 might contain a typo, as Synthcity, to the best of my knowledge, also supports qualitative metrics like t-SNE visualization. I also suggest to mention SDV in Table 1
>
> Thank you for bringing this to our attention. We will address the issue with the qualitative metrics. As for SDV, we listed DeepEcho in the table, as it is the SDV component for time series generation.
>
> > the choice of Keras is not adequately discussed or justified; it is unclear whether it is faster or slower compared to other libraries, especially since most recent models are available in PyTorch.
>
> There are two main reasons why we chose to use Keras:
> - **Convenient API for High-Level Model Definition**: Keras provides a detailed and flexible API that is ideal for creating unified yet customizable models for synthetic time series. This includes the easy integration of different losses and base architectures.
> - **Customizable to Different Backends**: In designing our framework, we aimed for compatibility with multiple backends. Keras's roadmap indicated a transition to supporting various backends, which was realized with the introduction of Keras 3. Keras 3 now supports PyTorch and JAX backends. We are currently aiming to migrate our framework to Keras 3, enabling the TSGM to support these backends as well (see [Issue #40](https://github.com/AlexanderVNikitin/tsgm/issues/40), [PR#50](https://github.com/AlexanderVNikitin/tsgm/pull/50)). The TSGM community is actively contributing to this transition.
> We will include comments on this in the next revision.
>
> > 2) the framework does not yet support more recent solutions, such as diffusion models or normalizing flows for time-series. While they could be integrated, the use of Keras might slow this integration [...] these solutions are mostly developed in PyTorch
>
> We completely agree with this point! Our efforts to enable multiple backends for TSGM are specifically aimed at addressing this issue. By defining only the architecture in Keras and reusing training loops from PyTorch, TSGM will be able to quickly integrate methods from other backends, such as JAX and PyTorch.
>
> We've added DDPM to the framework ([PR#56](https://github.com/AlexanderVNikitin/tsgm/pull/56)), and will support normalizing flows. Additionally, we will include a discussion on transformer-based and diffusion-based methods in the related work section.
>
> > improve writing, motivation, and discussion (e.g., why Keras).
>
> We will improve our writing according to your specific suggestions from the above.
>
> > introduce the predictive score metric [1] : how useful the synthetic data are for a downstream task in the generation case
>
> We evaluate utility in Fig. 4, and Tab. 2. For instance, `metrics.DownstreamPerformanceMetric` is used to evaluate predictive score (LL 161-165). We will expand on this.
>
> > discuss better the supported models, to give a bit more space (maybe appendix) to simulation, and how to find their parameters.
>
> We will address this by including a discussion on the applications of simulators in scientific fields such as astronomy [1], particle physics [2], econometrics [3], and population biology [4]. Furthermore, we will provide an explanation of how the parameters can be determined.
>
> **Additional References**
>
> [1] Dax, et al. (2021). Real-time gravitational wave science with neural posterior estimation.
> [2] Brehmer, et al. (2022). Simulation-based inference methods for particle physics.
> [3] Mariano, et al. (2000). Simulation-based inference in econometrics: Methods and applications.
> [4] Toni, et al. (2010). Simulation-based model selection for dynamical systems in systems and population biology.
>
> **Concluding remarks**
> We would be grateful if you could let us know whether our explanations have addressed your concerns. Please let us know if you have any other questions or concerns.

---

> > ### Comment · Reviewer_dX7D · 2024-08-20
> >
> > The authors have mostly addressed my concerns. I increase the score.

---

### Author Rebuttal · Authors · 2024-08-17

We thank all reviewers for their thoughtful reviews, valuable suggestions, and for taking the time to read our paper.

We particularly appreciate the positive recognition of many aspects of our work, including its clarity (dX7D, 6muW, S8WW), originality (6muW), significance (dX7D, BaRA, 6muW, S8WW), quality (dX7D, 6muW), and usefulness (BaRA, S8WW).
We hope we have addressed all questions and concerns raised by the reviewers and are happy to discuss any remaining concerns or questions during the rebuttal.

**Main Changes**
- Improving the clarity of the text with minor fixes and additional explanations,
- Adding a realistic dataset with regime changes MIT-BIH Arrhythmia Database ([PR#54](https://github.com/AlexanderVNikitin/tsgm/pull/54)),
- Implementing DDPM ([PR#56](https://github.com/AlexanderVNikitin/tsgm/pull/56)),
- Taking requested features into account and creating several Issues on GitHub for anticipated features development: normalizing flow ([Issue#55](https://github.com/AlexanderVNikitin/tsgm/issues/55)), Diffusion-TS ([Issue#57](https://github.com/AlexanderVNikitin/tsgm/issues/57)), TTS-GAN ([Issue#58](https://github.com/AlexanderVNikitin/tsgm/issues/58)), and creating a library of synthetic datasets ([Issue#53](https://github.com/AlexanderVNikitin/tsgm/issues/53)).

We would be grateful if you could let us know whether our explanations have satisfactorily addressed your concerns. We are also open to discussing any other questions you may have.

---

### Decision · Program_Chairs · 2024-09-26

**Decision:**

Accept (Poster)

**Comment:**

The paper introduces a general framework for time-series generative models, TSGM, which offers a variety of models (probabilistic models, simulation-based, etc) and different evaluation metrics. All reviewers agree that the paper is well-written and makes solid contribution to the community. There is limited novel contribution in new insights in machine learning. In addition, some recent development in the area, such as diffusion based models, are not included. The authors are encouraged to address these issues in the final version.